# LLM-CODEBOOK FOR EXTREME COMPRESSION OF LARGE LANGUAGE MODELS

## ABSTRACT

Large Language Models (LLMs) have exhibited outstanding performance in both understanding and generating language. However, their remarkable abilities often correlate with large model sizes, leading to challenges during deployment, inference, and training phases. While weight quantization and pruning are prevalent strategies, they tend to lose crucial information under extreme compression. In this paper, we propose LLM-Codebook for extreme compression of large language models (LLM-Codebook), which maps expansive LLMs (in GB) to compact codebooks (in KB). The foundation of LLM-Codebook is our novel Hessian-aware K-means algorithm, which clusters weights into codebooks based on Hessian information, preserving parameters that have significant impacts on predictions. Simultaneously, the tuning technique, LoRA is adopted to update layers that have not been compressed, aiming to recover performance using only a limited corpus. LLM-Codebook effectively preserves the generation and multi-task solving abilities of LLMs, surpassing advanced methods such as GPTQ, QLoRA, LLM-Pruner, and SparseGPT. We validate our approach by extremely compressing LLaMA-7B and Vicuna-7B to a memory requirement of 2GB (a 6x compression factor) while retaining 99% of the baseline performance. Furthermore, our approach maintains reasonable accuracy even under extreme compression ratio, achieving 90% of the original performance (36% better than GPTQ) when the model size is compressed to one-eighth.

## 1 INTRODUCTION

Recently, advanced language models, also known as Large Language Models (LLMs) (OpenAI (2023), Touvron et al., Thoppilan et al. (2022), Scao et al. (2022), Zeng et al. (2022), Touvron et al.), have showcased impressive capabilities in complex language modeling tasks. Yet, in spite of their stellar achievements, the computational and storage costs of LLMs raise substantial challenges. For example, GPT3-175B has a hundred billion parameters and requires multi-GPU to inference Zhang et al. (2022), which poses significant challenges to applying LLM models on mobile devices.

Conventional weight pruning and quantization methods have been widely used to reduce the significant amount of memory required by LLMs, but they are challenging under extreme compression conditions. The main reason for this is that they are lossy compression methods. Although the performance of the model can be preserved at a low compression ratio by discarding non-critical information, essential information is also lost at an extreme compression ratio, leading to a substantial decrease in model accuracy.

Structured pruning removes channels containing crucial information, leading to a significant decline in accuracy during extreme compression. Unstructured pruning is hardware-unfriendly and requires a large amount of indexing, making extreme compression unachievable. Quantization, at the point of extreme compression, suffers from inadequate expressive capacity leading to a significant performance degradation. Additionally, specialized hardware systems are required to support the operation of specific bit quantization, rendering the compressed models incompatible with older hardware systems.

To break the limitations of conventional methods, new compression methods must be designed, which can not only achieve a very high compression ratio but also represent rich information to minimize the loss of model accuracy. As illustrated in Figure 1, the weight has structural properties,

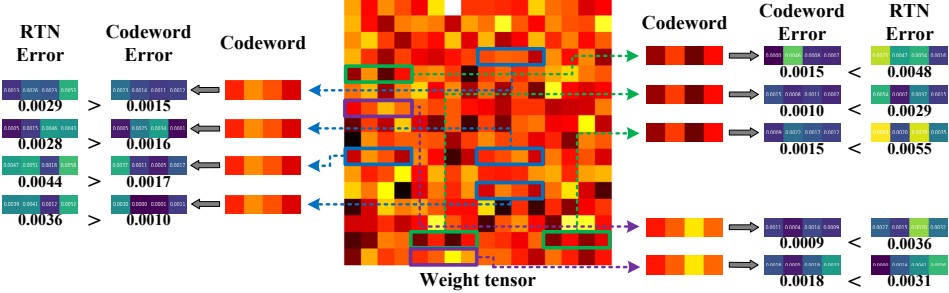

Figure 1: Illustration of LLM-Codebook. Compressing the V linear projection of the second transformer module of LLaMA-7B at 87.5% compression ratio compares LLM-Codebook with Round-To-Nearest (RTN). **RTN**: selects group-wise quantization granularity and rounds weight tensor to the nearest quantization level. **LLM-Codebook**: considers the vector repeatability of weight tensor and generates codewords of codebook to reconstruct vectors of weight tensor, which have much lower compression error than RTN.

and the structure weight-sharing strategy can reduce the weight compression error more effectively than the low-bit quantization method under extreme compression. Therefore, we propose LLM-Codebook for extreme compression of large language models (LLM-Codebook). To the best of our knowledge, it is the first time employing codebooks for LLMs compression. To ensure that the codebooks retain useful parameters for model prediction, LLM-Codebook is generated based on our proposed Hessian-aware K-means algorithm, which leverages Hessian information to identify and preserve salient parameters while clustering weights into codebooks. Meanwhile, Low-Rank Adaptation (LoRA) Hu et al. (2021) is employed to update the remaining uncompressed layers, with the objective of improving performance through a limited dataset.

LLM-Codebook is an automated compression framework wherein all codebooks are automatically generated, removing the necessity for manual design and completing in three hours and reducing the memory requirements to less than a quarter of the original. The compression process relies on a mere 50k corpus, reducing dependency on extensive fine-tuning datasets. Moreover, the compressed language model retains its proficiency in addressing a broad range of language tasks effectively.

Experiments show that LLM-Codebook outperforms existing work, i.e., LLaMA-7B and Vicuna-7B, on various tasks including language modeling and common sense QA. At a compression ratio of around 81%, LLM-Codebook can maintain 99% of its original performance. In contrast, GPTQ suffers a significant loss in accuracy, while LLM-Prunner and SparseGPT completely collapse. Under extreme compression conditions when the compression ratio exceeds 87%, LLM-Codebook can maintain about 90% of its original performance, while GPTQ collapses due to excessive weight quantization error.

## 2 RELATED WORK

**Large Language Model Compression.** Though LLMs have shown great potential in semantic understanding and language generation, their large model size greatly limits their application on mobile devices. Weight pruning and quantization are the two most commonly used methods to compress LLMs. For LLM pruning, LLM-Prunner Ma et al. (2023) computes the importance of channel-wise weights to perform structured pruning on the model and then fine-tunes the pruned model using LoRA (Hu et al., 2021). However, each channel contains crucial information, and pruning them can significantly degrade the performance. SparseGPT Frantar & Alistarh (2023) performs unstructured pruning on the weights while compensating for the weights that are not pruned. Similarly, the loss of important information cannot be compensated by updating the unpruned weights. In the field of LLM quantization, methods like ZeroQuant Yao et al. (2022), LLM.int8() Dettmers et al. (2022), LUT-GEMM Park et al. (2022) have shown the potential of 8-bit weight quantization, but they cannot be adopted in extreme compression condition. GPTQ Frantar et al. (2022) demonstrated efficient quantization of weights to 2-4 bits, which compensates for the weights based on the Hessian matrix. Since GPTQ employs RTN quantization, it suffers severe accuracy loss under extreme low-bit con-

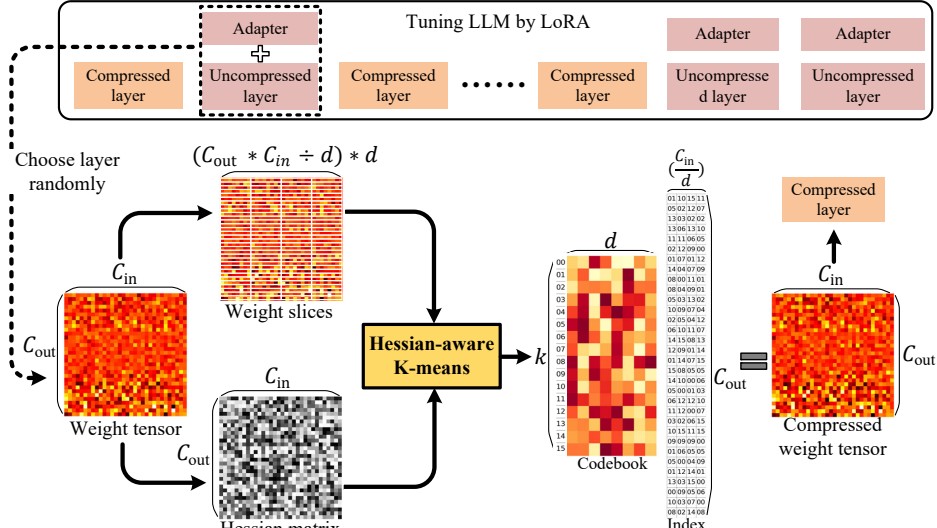

Figure 2: LLM-Codebook compression procedure. When LLM is being fine-tuned by LoRA, LLM-Codebook chooses the uncompressed layer randomly to compress, and then Hessian information can make codewords retain the salient parameters that contribute most to the prediction during the Hessian-aware K-means algorithm. Its weight tensor of size $C_{out} \times C_{in}$ is compressed by using a codebook of size $k \times d$ and index of size $\frac{C_{in}}{d} \times C_{out}$, which have low memory needs. Finally, the codebook and index are used to reconstruct the compressed weight tensor for the compressed layer.

ditions due to significant quantization errors of crucial information. QLoRA Dettmers et al. (2023) doubly quantizes the weights into 4-bit NormalFloat format and then fine-tunes them using LoRA, which faces the same extreme low-bit quantization issues.

**Compression based on Clustering.** Clustering is another weight compression method that reduces the model size by sharing weights. Unstructured weight-sharing methods like Deep Compression Han et al. (2015) quantize the weights to enforce weight sharing by K-means and adopt Huffman coding to encode indexes. However, as each parameter has a quantization index, these methods are not suitable for extreme compression. Some works focus on sharing weights structurally: Son et al. (2018) compresses CNNs by applying K-means clustering to convolution kernels so that redundancies are removed by sharing weights between similar kernels; Stock et al. (2019) uses the Expectation–Maximization algorithm to ensure that the output of each compressed layer remains the same as the original output and generates codebooks by K-means. They have not emphasized the retention of important information during the clustering process, leading to a decline in performance.

## 3 OUR APPROACH

As shown in Figure 2, to minimize the error caused by LLM-codebook compression, we choose layer-wise compression. Therefore, each uncompressed layer is randomly selected for compression, and then the following three stages are repeated until all uncompressed layers have been compressed: (1) The Salience Stage (Section 3.1) derives the salience of the random layer's weight through the Hessian matrix. (2) The Cluster Stage (Section 3.2) employs the Hessian-aware K-means algorithm to cluster the codebook which is then harnessed to restore the layer's weight. (3) The Recover Stage (Section 3.3) engages in a rapid tuning process to update the uncompressed layers so that performance drop can be mostly counteracted.

### 3.1 SALIENCE STAGE

We consider i-th fully-connected layer of LLM with weight $W_i \in \mathbf{R}^{C_{in} \times C_{out}}$. Our compression objective is to retain the parameters that have the most substantial influence on the model's predictive

performance, which is indicated by the deviation in loss and calculated using a public dataset $\mathcal{D} = \{x_m, y_m\}_{m=1}^n$ comprising **n** samples. Specifically, to estimate the salience of $W_i$, the deviation in the loss function caused by $W_i$ from zero to the current value can be formulated as (LeCun et al., 1989):

$$S_i = |\Delta\mathcal{L}(W_i; \mathcal{D})| = \left| \frac{\partial\mathcal{L}(\mathcal{D})}{\partial W_i}^\top W_i + \frac{1}{2}W_i^\top H W_i + \mathcal{O}\left(\|W_i\|^3\right) \right|, \quad \frac{\partial\mathcal{L}(\mathcal{D})}{\partial W_i}^\top \not\approx 0 \quad (1)$$

where $H$ is the Hessian matrix and $\mathcal{L}$ denotes the Cross-entropy loss: $\mathcal{L} = -\sum_m \log\left(p\left(y_m \mid x_m\right)\right)$, when $y_m$ is one-hot encoded. As Ma et al. (2023) describes, since $\mathcal{D}$ here is not a part of the original training data, the first term does not approach zero, thus it cannot be neglected. Then we derive the above function at a finer granularity, where each parameter $W_i^k$ within $W_i$ is calculated for its salience:

$$S_i^k = |\Delta\mathcal{L}(W_i^k; \mathcal{D})| = \left| \frac{\partial\mathcal{L}(\mathcal{D})}{\partial W_i^k} W_i^k + \frac{1}{2}W_i^k H_{kk} W_i^k + \mathcal{O}\left(\|W_i^k\|^3\right) \right| \quad (2)$$

where $k$ represents the k-th parameter in $W_i$ and $H_{kk}$ is the diagonal of $H$. However, the formula cannot be directly computed, since the computation of $H$ on the LLM is impractical due to its $\mathcal{O}\left(N^2\right)$ complexity. Thus, we demonstrate how to approximate $H$. According to Barshan et al. (2020), the Fisher information matrix $F$ of a conditional distribution parameterized by $p\left(y \mid x_m\right)$ is:

$$F = \frac{1}{n}\sum_{m=1}^n \mathbb{E}_p \nabla(\log p\left(y \mid x_m\right))\nabla(\log p\left(y \mid x_m\right))^T = -\frac{1}{n}\sum_{m=1}^n \mathbb{E}_p \nabla^2 \log p\left(y \mid x_m\right) \quad (3)$$

We assume that the pre-trained LLM has learned a distribution $p\left(y \mid x\right)$ close to the "true" distribution, such that $F$ is approximated by replacing $\mathbb{E}_p$ with a Monte Carlo estimate based on the target values $y_m$ in the public set:

$$F \approx -\frac{1}{n}\sum_{m=1}^n \nabla^2 \log p\left(y_m \mid x_m\right) = H \quad (4)$$

Therefore, $H$ can be replaced by $F$ in the computation of parameter salience:

$$S_i^k \approx \left| \frac{\partial\mathcal{L}(\mathcal{D})}{\partial W_i^k} + \frac{1}{2}W_i^k F_{kk} W_i^k + \mathcal{O}\left(\|W_i^k\|^3\right) \right| \approx \left| \frac{\partial\mathcal{L}(\mathcal{D})}{\partial W_i^k}W_i^k - \frac{1}{2}\left(\frac{1}{n}\sum_{m=1}^n \frac{\partial\mathcal{L}(\mathcal{D}_n)}{\partial W_i^k}W_i^k\right)^2 \right| \quad (5)$$

where the remainder term can be neglected. By utilizing any $S_i^k$, we estimate the salience at the granularity of each parameter of weight.

### 3.2 CLUSTERING STAGE

The clustering stage generates a codebook to reconstruct $W_i$. Each row of $W_i$ is split into $p$ contiguous subvectors and learns a codebook on the resulting $p \times C_{\text{out}}$ subvectors. Then, each subvector is mapped to its nearest codeword in this codebook.

For simplicity, we assume that $C_{\text{in}}$ is a multiple of $p$, and thus all subvectors have the same dimension $d = C_{\text{in}}/p$. The codebook $C_i = \{C_{i1}, \ldots, C_{ik}\}$ contains $k$ codewords of dimension $d$.

The following four steps explain how the codebook is learned by using the Hessian-aware K-means algorithm:

(1) **Setting the objective function.** The function is expressed as the squared Euclidean norm of the difference between original weight $W_i$ and compressed weight $\widehat{W_i}$ reconstructed by codebook :

$$\|W_i - \widehat{W_i}\|_2^2 = \sum_n \left\|W_{im} - \widehat{W_{im}}\right\|_2^2 = \sum_n \|W_{im} - c_{im}\|_2^2 \quad (6)$$

where $W_{im}$ represents the m-th subvector of $W_i$. $W_i$ is mapped to its compressed version $\widehat{W_i} = (c_{i1}, \ldots, c_{im})$ where $i1$ denotes the index of the codeword assigned to the first subvector $W_{i1}$, and so forth.

(2) **Initializing Codebook.** The saliences of subvectors of $W_i$ are computed in summation of $d$ parameters in each subvector: $S_{im} = \sum_{n=1}^{d} S_i^n$. The $k$ subvectors with the highest salience serve as the initial codewords in the codebook.

(3) **Assigning Clusters.** Each subvector $W_{im}$ is assigned to the nearest codeword by calculating the Euclidean distance from itself to each codeword $c_{ik}$:

$$\hat{k} = \arg\min_k \|W_{im} - c_{ik}\|_2^2 \tag{7}$$

where $\hat{k}$ represents the index of the nearest codeword to the subvector $W_{im}$.

(4) **Updating Codebook.** For each cluster where the subvectors have the same $\hat{k}$, the codeword is updated by using the salience-weighted mean of the subvectors in the cluster:

$$c_{ik} = \frac{\sum_{m \in cluster_{\hat{k}}} S_{im} \times W_{im}}{\sum_{m \in cluster_{\hat{k}}} S_{im} + \epsilon} \tag{8}$$

where $\epsilon$ is set to a small value (e.g., $1e{-}10$) to prevent division by zero.

Steps (3) and (4) are repeated until the number of updated codewords falls below the threshold or the number of iterations exceeds a specified limit, indicating that the codebook is completed.

### 3.3 RECOVER STAGE

In the recovery stage, we adopt the LoRA method to fine-tune the uncompressed layers so that the performance drop can be mostly counteracted. Given an original weight of uncompressed layer $W_j^o \in \mathbf{R}^{C_{in} \times C_{out}}$, LoRA modifies its update by using a low-rank decomposition represented as $W_j = W_j^o + \Delta W = W_j^o + BA$, where $B \in \mathbf{R}^{C_{in} \times r}$, $A \in \mathbf{R}^{r \times C_{out}}$, and the rank $r \ll \min(C_{in}, C_{out})$. During the fine-tuning process, $W_j^o$ remains constant without gradient updates, whereas $A$ and $B$ contain trainable parameters. Given a projection $Y = XW_j$ with $X \in R^{b \times C_{in}}$, the computation in this context can be expressed as:

$$Y = W_j X = (W_j^o + \Delta W)X = (W_j^o X) + s(BA)X \tag{9}$$

where $s$ is a scalar. The overall fine-tuning complexity is very low by optimizing $A$ and $B$. Additionally, the pre-trained weights of the uncompressed layer and the adapter are merged before compression to ensure that no parameters are added in the final compressed model.

## 4 EXPERIMENTS

**Overview.** We begin our experiments by validating the accuracy of LLM-Codebook relative to other lossy quantization and pruning methods on models with 7 billion parameters that provide acceptable runtimes. Next, we present the results of size compression ratios ranging from 75% to 95% for LLaMA-7B and Vicuna-7B models. The perplexity (PPL) and accuracy metrics are used for challenging language generation tasks and classification tasks, respectively. Thereafter, we show that LLM-Codebook remains stable under extreme compression ratio (compression ratio greater than 85%), whereas other compression methods have already collapsed. To complement this performance analysis, we also conduct metrics statistics for LLaMA-7B, including the number of parameters, memory needs, and compression ratio. In addition, we focus on investigating the Hessian-aware strategies and tuning strategies for the impact on LLM-Codebook, and conduct detailed ablation experiments. Finally, we employ the compressed model for text generation, demonstrating that the compressed model still maintains excellent performance.

**Setup.** We apply the LLM-Codebook compression method to the pre-trained LLaMA Touvron et al. (2023) and Vicuna Chiang et al. (2023) provided by Huggingface. Each model is compressed on a single 48 GB NVIDIA A6000 GPU for about 3 hours. To gauge our model's efficacy in a task-neutral context, we adopt the evaluation metrics that LLaMa uses for zero-shot task classification on common sense reasoning datasets: BoolQ Clark et al. (2019), PIQA Bisk et al. (2020), HellaSwag Zellers et al. (2019), WinoGrande Sakaguchi et al. (2021), ARC-easy Clark et al. (2018), ARC-challenge Clark et al. (2018), and OpenbookQA Mihaylov et al. (2018). As suggested by Gao

Table 1: Zero-shot performance of the compressed LLaMA-7B and Vicuna-7B. The average value is calculated among seven classification datasets. 'Bold' indicates the best performance within each 5% compression ratio interval. 'Ratio' denotes compression ratio. 'NF4-DQ' denotes that QLoRA uses a 4-bit NormalFloat format with double quantization. 'g128' denotes that RTN and GPTQ adopt group-wise quantization with a group size of 128. $(\mathbf{d}, \mathbf{k})$ represents each codebook has $k$ codewords of dimension $d$.

| Method | Ratio | Wiki2↓ | PTB↓ | ARC-c | ARC-e | BoolQ | HellaS | OBQA | PIQA | WinoG | Avg. |
|---|---|---|---|---|---|---|---|---|---|---|---|
| LLaMA-7B | 0.0% | 12.6 | 53.8 | 44.8 | 72.9 | 75.1 | 76.2 | 44.4 | 79.2 | 69.9 | 66.1 |
| QLoRA-NF4-DQ | 71.6% | 15.8 | 70.9 | 44.4 | 71.0 | 75.4 | **77.0** | 43.2 | 73.2 | 68.9 | 64.7 |
| RTN-4bit-g128 | 74.1% | 15.7 | 62.1 | 43.1 | 70.8 | 73.4 | 74.2 | 43.2 | 78.3 | 69.2 | 64.6 |
| GPTQ-4bit-g128 | 74.1% | **13.1** | **53.2** | 44.2 | 71.9 | 74.9 | 75.8 | **44.2** | 78.8 | 69.1 | 65.6 |
| SparseGPT | 75.0% | 107.7 | 358.8 | 23.3 | 34.0 | 60.9 | 33.8 | 26.0 | 56.6 | 53.3 | 41.1 |
| LLM-Pruner | 75.2% | 117.4 | 379.0 | 24.7 | 33.0 | 49.6 | 32.8 | 28.6 | 62.2 | 56.1 | 41.0 |
| $(\mathbf{d}, \mathbf{k}) = (4, 2^{15})$ | 75.9% | 13.4 | 56.0 | **46.1** | **73.2** | **75.8** | 75.5 | 43.4 | **80.0** | **70.1** | **66.3** |
| RTN-3bit-g128 | 80.3% | 90.0 | 368.8 | 27.4 | 45.3 | 48.0 | 45.5 | 29.2 | 65.8 | 53.5 | 45.0 |
| GPTQ-3bit-g128 | 80.3% | 15.0 | **60.4** | 42.2 | 69.4 | 70.7 | 72.5 | 41.8 | 77.8 | 67.1 | 63.0 |
| $(\mathbf{d}, \mathbf{k}) = (4, 2^{12})$ | 81.0% | **14.6** | 63.1 | **43.3** | 70.6 | **77.4** | 74.5 | 43.8 | 79.1 | **68.3** | **65.3** |
| SparseGPT | 82.5% | 443.1 | 1e3 | 24.2 | 28.7 | 39.0 | 29.1 | 24.2 | 52.7 | 49.5 | 35.3 |
| LLM-Pruner | 82.9% | 164.3 | 555.7 | 24.8 | 30.0 | 60.9 | 26.7 | 30.6 | 53.1 | 48.9 | 39.3 |
| RTN-2bit-g128 | 86.6% | 8e4 | 7e4 | 29.0 | 25.0 | 50.3 | 26.5 | 27.2 | 49.1 | 50.0 | 36.7 |
| GPTQ-2bit-g128 | 86.6% | 736.2 | 1e3 | 24.9 | 28.2 | 39.9 | 26.9 | 25.0 | 52.1 | 51.3 | 35.5 |
| $(\mathbf{d}, \mathbf{k}) = (8, 2^{15})$ | 87.1% | **20.2** | **80.7** | **37.2** | **63.2** | **70.7** | **66.2** | **38.2** | **73.0** | **63.4** | **58.8** |
| SparseGPT | 87.5% | 2e3 | 4e3 | 26.0 | 26.3 | 38.2 | 26.6 | 23.8 | 50.4 | 48.9 | 34.3 |
| LLM-Pruner | 87.9% | 1e4 | 1e4 | 26.9 | 26.9 | 40.0 | 25.7 | 26.8 | 49.5 | 52.2 | 35.4 |
| SparseGPT | 90.0% | 3e3 | 5e3 | 27.0 | 26.4 | 37.8 | 26.0 | 23.6 | 50.8 | 49.7 | 34.5 |
| LLM-Pruner | 90.0% | 2e4 | 1e4 | 27.5 | 27.2 | 41.1 | 25.5 | 29.2 | 49.0 | 49.0 | 35.4 |
| $(\mathbf{d}, \mathbf{k}) = (8, 2^{12})$ | 90.0% | **31.5** | **131.0** | **30.1** | **50.0** | **64.5** | **55.7** | **31.2** | **71.3** | **57.1** | **51.4** |
| $(\mathbf{d}, \mathbf{k}) = (16, 2^{15})$ | 91.9% | 121.5 | 380.5 | 24.3 | 36.7 | 39.7 | 32.8 | 27.6 | 58.0 | 52.5 | 38.8 |
| $(\mathbf{d}, \mathbf{k}) = (16, 2^{12})$ | 94.7% | 2e3 | 3e3 | 26.5 | 28.0 | 37.9 | 26.7 | 25.6 | 50.1 | 51.1 | 35.1 |
| Vicuna-7B | 0.0% | 17.1 | 63.1 | 44.5 | 71.9 | 78.1 | 73.9 | 43.8 | 79.1 | 69.1 | 65.8 |
| QLoRA-NF4-DQ | 71.6% | 17.3 | 67.0 | 42.5 | **71.1** | 77.2 | 73.6 | 42.0 | 78.5 | 68.4 | 64.8 |
| RTN-4bit-g128 | 74.1% | 21.5 | 74.0 | 42.7 | 68.6 | 71.0 | 71.4 | 42.8 | 77.9 | 65.4 | 62.8 |
| GPTQ-4bit-g128 | 74.1% | 17.8 | 65.1 | 41.7 | 70.8 | 75.8 | 73.6 | 42.2 | 78.5 | **70.1** | 64.7 |
| SparseGPT | 75.0% | 116.5 | 302.2 | 22.6 | 34.3 | 45.7 | 34.4 | 26.0 | 56.3 | 53.9 | 39.0 |
| LLM-Pruner | 75.2% | 140.5 | 479.1 | 25.4 | 30.9 | 62.2 | 27.3 | 31.4 | 54.6 | 49.5 | 40.2 |
| $(\mathbf{d}, \mathbf{k}) = (4, 2^{15})$ | 75.9% | **15.7** | **58.6** | **43.4** | 69.5 | **78.5** | 74.4 | **43.8** | 78.6 | 68.6 | **65.3** |
| RTN-3bit-g128 | 80.3% | 182.6 | 450.1 | 24.8 | 37.4 | 53.9 | 40.0 | 30.4 | 59.5 | 51.8 | 42.5 |
| GPTQ-3bit-g128 | 80.3% | 21.3 | 75.8 | 41.0 | 66.5 | 66.2 | 70.2 | 38.8 | 76.2 | 66.2 | 60.7 |
| $(\mathbf{d}, \mathbf{k}) = (4, 2^{12})$ | 81.0% | **16.2** | **59.5** | **41.2** | **67.7** | **78.4** | 72.3 | 42.4 | 78.2 | 67.9 | **64.0** |
| SparseGPT | 82.5% | 479.1 | 1e3 | 22.2 | 28.5 | 37.8 | 28.6 | 23.4 | 52.2 | 49.9 | 34.7 |
| LLM-Pruner | 82.5% | 210.1 | 831.0 | 26.0 | 30.0 | 60.7 | 26.8 | 31.2 | 53.3 | 49.7 | 39.7 |
| RTN-2bit-g128 | 86.6% | 1e5 | 1e5 | 28.0 | 26.7 | 50.0 | 26.1 | 29.0 | 47.8 | 48.4 | 36.6 |
| GPTQ-2bit-g128 | 86.6% | 6e3 | 7e3 | 25.8 | 25.9 | 44.6 | 26.4 | 27.4 | 50.9 | 52.0 | 36.1 |
| $(\mathbf{d}, \mathbf{k}) = (8, 2^{15})$ | 87.1% | **21.0** | **71.8** | **35.1** | **60.0** | **72.0** | **65.2** | **37.0** | **74.8** | **63.7** | **58.2** |
| SparseGPT | 87.5% | 2e3 | 4e3 | 26.5 | 26.4 | 37.8 | 26.0 | 23.4 | 50.4 | 49.3 | 34.3 |
| LLM-Pruner | 87.9% | 7e3 | 6e3 | 26.7 | 26.5 | 46.2 | 25.0 | 26.8 | 49.0 | 49.3 | 35.6 |
| SparseGPT | 90.0% | 3e3 | 5e3 | 26.3 | 26.1 | 38.0 | 25.6 | 22.6 | 50.2 | 48.9 | 34.0 |
| LLM-Pruner | 90.0% | 7e3 | 5e3 | 27.0 | 26.1 | 39.2 | 24.8 | 27.4 | 49.6 | 48.8 | 34.7 |
| $(\mathbf{d}, \mathbf{k}) = (8, 2^{12})$ | 90.0% | **33.2** | **99.3** | **31.0** | **52.3** | **60.1** | **53.1** | **32.6** | **69.0** | **57.2** | **50.8** |
| $(\mathbf{d}, \mathbf{k}) = (16, 2^{15})$ | 91.9% | 91.8 | 286.1 | 23.6 | 37.3 | 58.4 | 34.5 | 23.2 | 59.9 | 51.1 | 41.1 |
| $(\mathbf{d}, \mathbf{k}) = (16, 2^{12})$ | 94.7% | 9e3 | 1e4 | 26.0 | 25.9 | 38.1 | 26.4 | 26.0 | 50.3 | 50.5 | 34.7 |

et al. (2021), our model either ranks the options in a multiple-choice task or generates open-ended answers. Furthermore, we also supplement the zero-shot perplexity evaluation metrics including Wikitext2 Merity et al. (2016) and PTB Marcus et al. (1993).

**Baselines.** Our primary baseline is the Round-To-Nearest (RTN) quantization, which has shown to be particularly effective for small group sizes, such as 128. We also benchmark against state-of-the-art methods such as GPTQ Frantar et al. (2022) for Post-Training Quantization (PTQ), QLoRA Dettmers et al. (2023) for Quantization-Aware Training (QAT), LLM-Prunner Ma et al. (2023) for structured pruning, and SparseGPT Frantar & Alistarh (2023) for unstructured pruning.

Table 2: Metrics of LLM-Codebook for LLaMA-7B. $(\mathbf{d}, \mathbf{k})$ represents each codebook has $k$ codewords of dimension $d$.

| | LLaMA-7B | $\left(4, 2^{15}\right)$ | $\left(4, 2^{12}\right)$ | $\left(8, 2^{15}\right)$ | $\left(8, 2^{12}\right)$ | $\left(16, 2^{15}\right)$ | $\left(16, 2^{12}\right)$ |
|---|---|---|---|---|---|---|---|
| #Params | 6.74B | 1.91B | 1.89B | 1.13B | 1.08B | 0.78B | 0.68B |
| Memory Needs | 12.38GB | 2.98GB | 2.35GB | 1.60GB | 1.24GB | 1.00GB | 0.66GB |
| Compression Ratio | 0.0% | 75.91% | 81.00% | 87.07% | 90.00% | 91.92% | 94.67% |

**Metrics.** Table 2 shows the metrics of the models employed in our experiments: the number of parameters, the memory requirement for each model's operation, and the compression ratio. The dimension and number of codewords in each codebook are set to $d \in \{4, 8, 16\}$ and $k \in \{2^{15}, 2^{12}\}$ for each compression regime, respectively. Furthermore, We do not compress the embedding layer and the output layer, as they are responsible for embedding the text or transforming the features into text. The memory requirements of these two layers are also not accounted for in the experiments. The memory requirement after compressing is calculated as the sum of the indexing cost (number of Byte per subvector of weight) and the overhead of storing the codebooks in bfloat16 format: $\frac{C_{\text{in}} \times C_{\text{out}}}{d} \times \frac{\log_2(k)}{8} B + k \times d \times 2B$. Therefore, increasing $d$ can reduce the compression ratio more significantly than decreasing $k$ when $C_{\text{in}}$ and $C_{\text{out}}$ have large values such as 4096. For instance, compressing weight of size $4096 \times 4096$ with $k = 2^{15} = 32768$ codewords (15/8 Byte for the index of each subvector) and a codeword dimension of $d = 8$ results in an indexing cost of $3.84$ MB for $m = 2,097,152$ subvectors, plus the cost of storing the codebook of $512$ kB. This reduces the memory requirement from 32MB to 4MB, achieving a compression ratio of about 87%.

**Layer sensitivity in Compression.** We conduct multiple experiments, and each time only one layer is compressed with a codebook size of $(\mathbf{d}, \mathbf{k}) = \left(16, 2^{12}\right)$. As depicted in figure 3, the transformer modules across different layers exhibit an uneven distribution of sensitivity, with the first three layers and the last two layers having a more important impact on the model's performance. In other words, compressing these layers suffers from more significant performance degradation than other layers. Moreover, the parameter count of each FFN linear projection is three times that of each QKVO linear projection in the same transformer layer. To address the above-mentioned two issues, we double the codebook size (number of codewords) of the QKVO linear projections and increase the codebook size of the FFN linear projections to four times in these layers. In other layers, we decrease the codebook size of the QKVO linear projections to half of its initial size and keep the overall codebook size of the FFN linear projections unchanged. This ensures a consistent overall compression ratio.

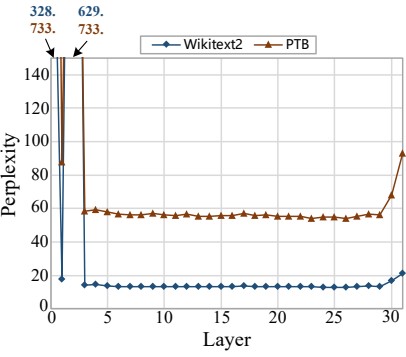

Figure 3: Layer sensitivity for Compressing: compressing weights in only one layer of LLaMA-7B. The perplexity of the first layer for Wikitext2 and PTB is 328 and 733, respectively, and the perplexity of the third layer is 629 and 733, respectively.

**Hyperparameters.** From Bookcorpus Zhu et al. (2015), we randomly select 15 samples and truncate each of them to a sequence length of 128 to compute the salience for both LLaMA and Vicuna. In the recovery stage, we utilize the clean version of Alpaca Taori et al. (2023), which contains approximately 50k samples. The batch size for fine-tuning is set to 64 and the AdamW optimizer is employed in our experiment. The learning rate, the LoRA rank, the LoRA alpha, and the LoRA dropout are set to 0.0001, 128, 1, and 0.05, respectively. For every $3 + m \times 9$ iterations, an uncompressed layer is randomly selected for compression, where $m$ denotes the ratio of the number of compressed layers to the number of all layers.

**Zero-shot Performance** Table 1 presents the zero-shot performance of the compressed model. Based on the evaluation conducted on LLaMA, the LLM-Codebook outperforms the advanced pruning methods and quantization methods by significant margins. For example, at a compression ratio

Table 3: Ablation study on LLaMA-7B. The data is obtained by subtracting the results of the complete method from that of the ablation method with the same codebook size. The average value is calculated among seven classification datasets. The 'Sal' option indicates whether salience is used in the Clustering Stage, while the 'Tune' option denotes whether LoRA is employed for fine-tuning during the Recovery Stage. $(\mathbf{d}, \mathbf{k})$ represents each codebook has $k$ codewords of dimension $d$.

| Method | Sal | Tune | Wiki2↓ | PTB↓ | ARC-c | ARC-e | BoolQ | HellaS | OBQA | PIQA | WinoG | Avg. |
|---|---|---|---|---|---|---|---|---|---|---|---|---|
| $(4, 2^{15})$ | | | +1.0 | +3.0 | -1.2 | -1.9 | -3.1 | -1.2 | -2.8 | -1.7 | -1.3 | -2.0 |
| $(4, 2^{15})$ | ✓ | | +0.4 | +0.7 | -2.4 | -2.0 | -3.0 | -1.7 | -2.6 | -1.4 | +0.7 | -1.8 |
| $(4, 2^{15})$ | | ✓ | +1.0 | +4.3 | -0.2 | +0.5 | -1.8 | -0.5 | -1.4 | -0.4 | +0.1 | -0.6 |
| $(4, 2^{12})$ | | | +5.6 | +15.2 | -4.2 | -5.1 | -4.3 | -6.2 | -1.0 | -2.3 | -4.9 | -4.0 |
| $(4, 2^{12})$ | ✓ | | +1.8 | +3.8 | -3.5 | -2.6 | -2.1 | -5.0 | -2.6 | -1.5 | -1.5 | -2.7 |
| $(4, 2^{12})$ | | ✓ | +1.5 | +3.6 | +0.3 | -0.1 | -3.3 | -1.3 | -0.6 | +0.3 | -2.5 | -1.0 |
| $(8, 2^{15})$ | | | +218.0 | +466.1 | -8.4 | -15.8 | -26.2 | -31.6 | -7.8 | -13.6 | -10.5 | -16.2 |
| $(8, 2^{15})$ | ✓ | | +76.7 | +179.0 | -7.1 | -11.3 | -18.0 | -28.2 | -6.2 | -9.2 | -10.9 | -12.9 |
| $(8, 2^{15})$ | | ✓ | +5.1 | +9.3 | -0.4 | +0.3 | -5.0 | -3.0 | -0.2 | -2.1 | -0.8 | -1.6 |
| $(8, 2^{12})$ | | | +1e4 | +2e4 | +1.7 | -12.9 | +9.9 | -10.6 | +2.0 | -12.8 | -1.5 | -3.5 |
| $(8, 2^{12})$ | ✓ | | +1e4 | +9e3 | -0.0 | -12.7 | -4.3 | -11.0 | -2.6 | -12.9 | -0.2 | -6.3 |
| $(8, 2^{12})$ | | ✓ | +301.3 | +1e4 | -1.4 | -7.2 | +13.9 | -5.2 | +0.2 | -7.1 | -1.1 | -1.2 |

of 75%, maintain the same level of performance as the original model in terms of accuracy and perplexity. The pruning methods LLM-Pruner and SparseGPT experience a 20% decline in accuracy, while the quantization methods QLoRA and GPTQ also see a 1-2% decrease in accuracy. Furthermore, when the model size is compressed to one-eighth, LLM-Codebook achieves 90% of the original performance, while other methods have already failed because they have lost some crucial information in such a high compression ratio. Specifically, LLM-Codebook still manages to maintain nearly 80% accuracy in classification tasks when the model is compressed to one-tenth of its original size. LLM-Codebook only collapses completely when the compression ratio reaches 95%. The compression results of Vicuna-7B align with those of LLaMA-7B, as compressing 87% of model size on Vicuna-7B maintains performance at about 90%, while other methods have already collapsed. These results validate the effectiveness of LLM-Codebook in extreme model compression.

**Ablation Study.** We conduct ablation experiments including whether to employ salience for clustering or whether to perform tuning, as shown in Table 3. To investigate the importance of salience in the Hessian-aware K-means algorithm, we use a normal K-means algorithm to compress weight only based on weight values. Without tuning by LoRA, the perplexity of the normal method is 3x higher than that of the Hessian-aware method, which means that the Hessian-aware method can retain the weight parameters that have the greatest influence on model predictions. In the case of employing the normal K-means algorithm, the perplexity and accuracy of fine-tuning with LoRA are better than that without fine-tuning. Especially, at a compression ratio of 90%, there is a decrease of more than 200 in perplexity. This implies that the updating information of the weights can effectively compensate for the performance drop, which is also encoded into the codebook.

**Compression error.** As illustrated in Figure 4, under equivalent compression ratios, LLM-Codebook consistently outperforms RTN quantization. The compression ratio of LLM-Codebook is 7% higher than that of RTN to achieve a close Mean Absolute Error (MAE). The smaller compression error enables LLM-Codebook to maintain acceptable performance under extreme compression scenarios, while RTN-based methods like GPTQ collapse due to the large quantization error of much crucial information. This suggests that there are a large number of similar vectors in the LLM weight parameters, which can be represented by the same codeword.

**Generation Results.** Table 4 provides generation results from the original models and the compressed models. The results from the compressed model outputs display coherence, pertinence, and insight in relation to the assigned topic, each of which aligns well with those from the original model. This indicates that the source of this information is more from the original model rather than the tuning dataset.

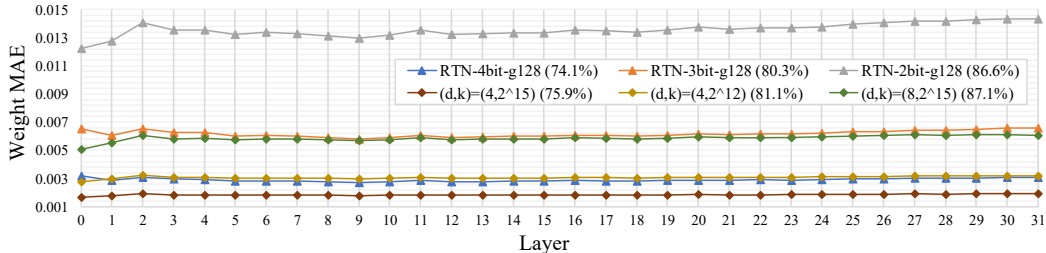

Figure 4: Mean Absolute Error (MAE) between uncompressed weights and weights compressed by Round-To-Nearest (RTN) quantization or LLM-Codebook in different layers of LLaMA-7B. The percentage in the legend represents the compression ratio. $(\mathbf{d}, \mathbf{k})$ represents each codebook has $k$ codewords of dimension $d$.

Table 4: Language generation results obtained with the original model and the compressed model. 'Ratio' denotes compression ratio.

| Ratio | LLaMA-7B | Vicuna-7B |
|---|---|---|
| 0.0% | List 5 reasons why someone should learn to code: 1. Increase marketability and job opportunities: Being able to write code in any language means you can create software, websites or apps that can be used for a variety of purposes. Having this skill is very valuable because it opens up doors to jobs in the tech industry... | The differences between genetic traditional and engineering breeding is the ability to manipulate genes at a molecular level, which is not possible with traditional breeding. Genetic engineering allows for the creation of new traits that are not found in nature. This can be used for many purposes... |
| 81.0% | List 5 reasons why someone should learn to code: 1. Increased Employment Opportunities – With the ever-growing demand for computer programmers, coders have an advantage in job opportunities compared to non-coding graduates. Many businesses now use technology to enhance their operations and require programmers... | The differences between genetic traditional and engineering breeding is the way in which genes are manipulated. With traditional breeding, breeders selectively choose traits to pass on from one generation to another through selection of individuals with desirable characteristics. In contrast... |
| 87.1% | List 5 reasons why someone should learn to code: 1. Career opportunities: One of the biggest reasons is that there will always be a demand for people who can code. Jobs in IT are growing rapidly and they're not going away anytime soon, making it a great time to start learning how to code. 2. Personal development... | The differences between genetic traditional and engineering breeding is that traditional breeding relies on the physical appearance of the animals to choose traits, while genetic engineering uses selective DNA sequencing for making the decisions. Traditional breeding focuses more on specific traits... |

## 5 CONCLUSION

In this work, we have proposed a LLM-Codebook for extreme compression of large language models (LLM-Codebook), effectively mapping extensive LLMs (in GB) to compact codebooks (in KB). Central to LLM-Codebook is our Hessian-aware K-means algorithm that clusters weights into codebooks based on Hessian information, ensuring the preservation of critical parameters impacting predictions. Moreover, we have leveraged tuning techniques LoRA to update uncompressed layers, aiming to restore performance utilizing only a limited corpus. LLM-Codebook has demonstrated notable preservation of generative and multi-task solving capabilities inherent to LLMs, surpassing conventional techniques like GPTQ, QLoRA, LLM-Pruner, and SparseGPT. Through rigorous evaluation, we have showcased the remarkable efficacy of LLM-Codebook by compressing LLaMA-7B and Vicuna-7B to a mere memory requirement of 2GB (a 6x compression factor), while maintaining 99% of the baseline performance. Notably, our method has upheld considerable accuracy under extreme compression ratios, attaining 90% of the original performance, a significant 36% improvement over GPTQ when the model size is compressed to one-eighth. The results affirm the potential of LLM-Codebook as a viable solution to the challenges of deployment, inference, and training associated with LLMs.

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
