# OpenReview forum: "LLM-Codebook for Extreme Compression of Large Language Models"
_ICLR.cc/2024/Conference — Submitted to ICLR 2024_

### Official Review · Reviewer_5akx · 2023-10-28

**Soundness:** 3 good
**Presentation:** 2 fair
**Contribution:** 2 fair
**Rating:** 6
**Confidence:** 4

**Summary:**

This paper proposes LLM-Codebook, an effective structure-clustering-based LLM compression technique for extreme compression. The main technology parts of this method consist of three steps: (1) Salient weight detection; (2) Hessian-aware K-means algorithm for weight clustering and compression; (3) Lora-based finetuning for retaining performance. The overall method is simple and effective. The manuscript is well-written with clear logic.

**Strengths:**

●	The paper is well-written and easy to follow.

●	The proposed LLM codebook shows good compression performance for a lower compression ratio as compared to recent compression works like GPTQ, SparseGPT, and LLM-Pruner.

**Weaknesses:**

●	It is unclear how can this method be combined with downstream LLM finetuning or if it is only effective for post-finetuning compression. After the compression, each linear layer consists of a codebook and an index map, how can the model be further finetuned under this structure?

●	Since the method requires full model finetuning using Lora for performance recovery, would the memory cost be huge when meeting with a larger base model like llama-70B?

●	It is still being determined why the layer is randomly selected for compression during the Lora finetuning procedure.

●	The compressed weight tensor in Figure. 2 seems like a copy of the original weight tensor, which looks too similar to be a real compressed visualization. Please explain this part.

●	Some statements in this paper related to existing compression methods are given lacking enough verification. The latest works are left without discussion. Taking the low-bit compression parts as an example, though the selected baseline GPTQ in this paper does not give a good performance for lower-bit like 2-bit, there are already several works showing promising results for lower-bit compression ([1-3]). For example, both omniquant and low_bit_llama show that llama families (1.1B-70B) can be well compressed to 2-bit with good performance. It is suggested to reorganize this part and discuss the possibility of combining low-bit compression with structure-clustering for further compression.

Overall, this paper presents a method for compressing LLMs using structure clustering. The algorithm is verified on recent small LLMs like LLaMA-7B and shows good compression performance as compared to some of the existing techniques. The overall method is simple and effective. Some statements in the paper are not well verified.

[1] omniquant: omnidirectionally calibrated quantization for large language models

[2] https://github.com/GreenBitAI/low_bit_llama

[3] QA-LoRA: Quantization-Aware Low-Rank Adaptation of Large Language Models

**Questions:**

Pls. see the weaknesses.

---

> ### Author Response · Authors · 2023-11-22
> **Official Comment 1**
>
> Dear reviewer,
>
> I apologize for the delayed response. Your insightful feedback has been immensely valuable in guiding the refinement of our work. Despite the constraints of limited laboratory resources, I have endeavored to conduct as many supplementary experiments as possible to address your comments comprehensively.
>
> ---
> The updated part of our method:
>
> Inspired by the approaches of AWQ[1] and SqueezeLLM[2] in preserving important weights, we upgrade Hessian-aware K-means algorithm to retain the weight and position of the salience value in the first 0.66% in a compressed sparse row. This significantly enhances the compression performance with a 2% reduction in the compression ratio. The preserved weights are excluded from the Hessian-aware k-means clustering, which also reduces clustering errors.
>
> ---
> A1: In the codebook-based LoRA during the recovery stage, the original weights of each linear layer are frozen, and an adaptor is assigned to each layer. During the fine-tuning process of all adaptors, a linear layer is randomly selected, and its original weight is merged with the adaptor for compression, resulting in a codebook and index map. Subsequently, the compressed weight is reconstructed based on these components. The compressed weight is no longer updated, meaning that the codebook and index map of that specific linear layer are also not updated. To compensate for the compression error of the layers that have already been compressed, the other linear layers that have not yet been compressed continue to update their adaptors. The above process is repeated until all the linear layer are completely compressed.
>
> ---
> A2: The memory cost is primarily divided into two components: the base model and the adaptor. Typically, the adaptor accounts for about 1 to 3% of the base model's memory requirement, depending on the chosen value of the rank parameter. The memory consumption of LLaMA-70B exceeds 130GB, and regrettably, we lack sufficient computational resources to support the compression of this model.
>
> ---
> A3: Intel's Incremental Network Quantization[3] adopts a progressive quantization strategy, which divides weights into two groups. One group is quantized and then frozen, while the other is fine-tuned to compensate for the quantization loss of the former. This process is repeated until all weights are fully quantized. Inspired by the aforementioned strategy, the codebook-based LoRA in the recovery stage also represents a form of progressive compression, with specific procedures as outlined in answer1.
>
> ---
> A4: I apologize for the inadequacy of our figures in clearly demonstrating the differences. In fact, there are distinctions between the two weight maps; for instance, the weights in the top right corner differ in color. The issue arises from our approach of using heatmaps based on the min/max values of the weights. Given that these min/max values are similar for both sets of weights, the RGB differences corresponding to minor compression errors may be challenging to discern.
>
> ---
> A5: In Table 1 of Official Comment 2, we have supplemented the low-bit compression comparison with Omniquant[4], SqueezeLLM[2], and AWQ[1], demonstrating that our work achieves optimal performance in 2/3-bit compression of LLaMA-7b and LLaMA-13B. We do not include low_bit_llama[5] in our comparison due to the absence of open-source compression code and the lack of pre-compressed LLaMA1 models. Although QA-LoRA[6] represents an excellent piece of work, the absence of open-source code precludes comparison. In exploring the potential of combining low-bit compression with structural clustering for further compression, we conducted experiments with Codebook PTQ. As shown in Table 2 of Official Comment 2, the results for W8A16 and W4A16 are very close to those of W16A16.
>
> ---
> Reference:
>
> [1] Lin J, Tang J, Tang H, et al. AWQ: Activation-aware Weight Quantization for LLM Compression and Acceleration[J]. arXiv preprint arXiv:2306.00978, 2023.
>
> [2] Kim S, Hooper C, Gholami A, et al. SqueezeLLM: Dense-and-Sparse Quantization[J]. arXiv preprint arXiv:2306.07629, 2023.
>
> [3] Zhou A, Yao A, Guo Y, et al. Incremental network quantization: Towards lossless cnns with low-precision weights[J]. arXiv preprint arXiv:1702.03044, 2017.
>
> [4] https://github.com/GreenBitAI/low_bit_llama
>
> [5] Xu Y, Xie L, Gu X, et al. QA-LoRA: Quantization-Aware Low-Rank Adaptation of Large Language Models[J]. arXiv preprint arXiv:2309.14717, 2023.

---

> ### Author Response · Authors · 2023-11-22
> **Official Comment 2**
>
> ### Table 1: Zero-shot performance of the compressed LLaMA-7B and LLaMA-13B. ‘rec’ indicates whether LLM-codebooks use the recovery stage. (𝐝,𝐤) represents each codebook has 𝐤 codewords of dimension 𝐝.
>
> | Ratio  | Method | Wiki2↓ (LLaMA-7B) | PTB↓ (LLaMA-7B) | Wiki2↓ (LLaMA-13B) | PTB↓ (LLaMA-13B) |
> | ------ | ------ | --------------------------- | -------------------------- | ----------------------------- | ---------------------------- |
> | 0.0%   | LLaMA-7B | 12.6 | 53.8 | 11.58 | 44.56 |
> | 73.3%  | SqueezeLLM-4bit | 12.82 | 54.81 | 11.73 | **45.04** |
> | 73.3%  | AWQ-128g-4bit | 12.96 | 56.56 | 11.94 | 45.05 |
> | 73.3%  | OmniQ-128g-4bit | 12.87 | 54.93 | **11.71** | 45.73 |
> | 73.9%  | (𝐝,𝐤)=(4,2^15) | 12.82 | 54.81 | 11.95 | 45.70 |
> | 73.9%  | (𝐝,𝐤)=(4,2^15)-rec | **12.81** | **54.78** | 11.94 | 45.68 |
> | 79.7%  | SqueezeLLM-3bit | 13.91 | 57.22 | 12.75 | 46.93 |
> | 79.7%  | AWQ-128g-3bit | 14.89 | 59.59 | 12.92 | 48.35 |
> | 79.7%  | OmniQ-128g-3bit | 13.78 | 57.67 | **12.36** | 48.27 |
> | 80.0%  | (𝐝,𝐤)=(4,2^12) | 13.59 | 56.11 | 12.62 | 47.16 |
> | 80.0%  | (𝐝,𝐤)=(4,2^12)-rec | **13.55** | **55.87** | 12.59 | **46.90** |
> | 85.8%  | OmniQ-128g-2bit | 22.73 | 111.24 | 17.88 | 72.75 |
> | 86.1%  | (𝐝,𝐤)=(8,2^15) | 23.48 | 88.97 | 18.50 | 67.42 |
> | 86.1%  | (𝐝,𝐤)=(8,2^15)-rec | **19.92** | **75.21** | **17.18** | **62.22** |
>
> ---
> ### Table 2: Combining low-bit post-training quantization (PTQ) with LLM-Codebook without recovery stage. ‘g128’ represents PTQ method with a group size of 128 is used for all the codebooks. (𝐝,𝐤) represents each codebook has 𝐤 codewords of dimension 𝐝.
> | Method | Recovery | Bits | Wiki2↓ (LLaMA-7B) | PTB↓ (LLaMA-7B) | Wiki2↓ (LLaMA-13B) | PTB↓ (LLaMA-13B) |
> | ------ | -------- | ---- | --------------------------- | --------------------------- | ---------------------------- | ---------------------------- |
> | (𝐝,𝐤)=(4,2^15)  | × | W16A16 | 12.82 | 54.81 | 11.95 | 45.70 |
> | (𝐝,𝐤)=(4,2^15), PTQ-g128 | × | W8A16 | 12.82 | 54.81 | 11.95 | 45.70 |
> | (𝐝,𝐤)=(4,2^15), PTQ-g128 | × | W4A16 | 13.46 | 57.22 | 11.95 | 45.70 |
> | (𝐝,𝐤)=(4,2^15), PTQ-g128 | × | W3A16 | 16.75 | 67.16 | 14.33 | 53.54 |
> | (𝐝,𝐤)=(4,2^12)  | × | W16A16 | 13.59 | 56.11 | 12.62 | 47.16 |
> | (𝐝,𝐤)=(4,2^12), PTQ-g128 | × | W8A16 | 13.59 | 56.11 | 12.62 | 47.16 |
> | (𝐝,𝐤)=(4,2^12), PTQ-g128 | × | W4A16 | 13.59 | 56.11 | 12.62 | 47.16 |
> | (𝐝,𝐤)=(4,2^12), PTQ-g128 | × | W3A16 | 18.11 | 73.18 | 15.37 | 56.55 |
> | (𝐝,𝐤)=(8,2^15) | × | W16A16 | 23.48 | 88.97 | 18.50 | 67.42 |
> | (𝐝,𝐤)=(8,2^15), PTQ-g128 | × | W8A16 | 23.48 | 88.97 | 18.50 | 67.42 |
> | (𝐝,𝐤)=(8,2^15), PTQ-g128 | × | W4A16 | 23.48 | 88.97 | 19.93 | 73.18 |
> | (𝐝,𝐤)=(8,2^15), PTQ-g128 | × | W3A16 | 44.39 | 192.81 | 27.78 | 109.86 |

---

### Official Review · Reviewer_kbsm · 2023-10-29

**Soundness:** 2 fair
**Presentation:** 3 good
**Contribution:** 2 fair
**Rating:** 3
**Confidence:** 4

**Summary:**

This paper proposed a model weights compression algorithm method based on Hessian-aware K-means, especially for extreme reduction of model size. The authors empirically demonstrate the efficacy of Hessian-aware K-means and Lora-based recovery stage in compression and performance maintenance.

**Strengths:**

- The paper proposed to adopt an importance-aware K-means for model weights compression.
- The paper is well-written and easy to follow.

**Weaknesses:**

- The experimental results are not very clear in terms of fair comparison with previous methods.
- To make the paper stronger, the authors should provide more insightful explanations to connect the components of the proposed method, otherwise can be easily understood as a simple combination of two mature methods.

**Questions:**

- Eq(5) implies that the estimation of model weight salience depends on the selected dataset and I noticed that the authors used only 15 randomly selected samples from the Bookcorpus dataset. I wonder how sensitive this estimation is in terms of 1) the data source. Eg, how about 15 samples from Wiki or a similar corpus? 2) the number of samples. Eg, is 15 samples enough for a good estimation of salience? What’s the possible influence of the salience estimation error on the later recovery stage?


- In Table 3, the Lora-based tuning looks more critical to prevent the performance from unacceptable degradation. This makes the readers very curious about several questions. 1) With only vanilla K-means, is it possible to match the best performance if more effort is put in tuning the recovery stage? 2) What’s the must-be reason to use Hessian-aware K-means instead of vanllina k-means given its extra computation cost of estimation and secondary role in recovering model performance?


- Following Q.2 above, what are the baselines in Table 1 that support Lora-based tuning? If an extra recovery stage based on Lora tuning was carefully added, will those baseline performances catch up with the proposed method?

- In Table 3 last row, The salience-adopted clustering stage leads to more performance degradation compared to the vanilla baseline: -3.5 vs - 6.3. Can the author explain the reason for this observation?

---

> ### Author Response · Authors · 2023-11-22
> **Official Comment 1**
>
> Dear reviewer,
>
> I apologize for the delayed response. Your insightful feedback has been immensely valuable in guiding the refinement of our work. Despite the constraints of limited laboratory resources, I have endeavored to conduct as many supplementary experiments as possible to address your comments comprehensively.
>
> ---
> The updated part of our method:
>
> Inspired by the approaches of AWQ[1] and SqueezeLLM[2] in preserving important weights, we upgrade Hessian-aware K-means algorithm to retain the weight and position of the salience value in the first 0.66% in a compressed sparse row. This significantly enhances the compression performance with a 2% reduction in the compression ratio. The preserved weights are excluded from the Hessian-aware k-means clustering, which also reduces clustering errors.
>
> ---
> A1: In Table 1 of Official Comment 2, we conducted comparative analyses focusing on the sources of data and the number of samples used. We found that data derived from the Bookcorpus dataset yielded more accurate salience calculations compared to those obtained from Wiki and the C4 corpus. Additionally, increasing the number of samples further enhanced the precision of salience estimation. As shown in Table 2 of Official Comment 2, we performed ablation experiments by omitting the use of salience. These experiments demonstrated that preserving weights with higher salience significantly improves compression performance in the recovery stage.
>
> ---
> A2: In Table 3 of Official Comment 2, we conducted comparative experiments with different LoRA ranks, attempting to achieve optimal performance using only the conventional K-means algorithm while minimally increasing the number of trainable parameters. The results indicate that merely employing the standard K-means algorithm does not suffice to attain optimal performance.
> Contrary to playing a secondary role, Hessian-aware K-means is significantly influential in model compression. In fact, its impact on compression performance exceeds that of the recovery stage at 4/3-bit compression. Conversely, the recovery stage has a more pronounced effect in 2-bit compression scenarios.
>
> ---
> A3: The QLoRA and LLM-Pruner are based on LoRA tuning, while GPTQ, SparseGPT, AWQ, SqueezeLLM, and Omniquant[3] represent Post-Training Quantization (PTQ) methods or pruning methods not based on LoRA. In Table 3 of Official Comment 2, we have provided data that excludes the use of a recovery stage, and the results surpass those of the aforementioned methods. However, incorporating LoRA into PTQ methods involves substantial engineering efforts, and the adaptor weights fine-tuned are in fp16 format, which does not qualify as strict low-bit quantization. It indicates that our codebook-based LoRA approach in the recovery stage marks a significant innovation. This method deviates from earlier practices in CNN-related research, where updating the codebook was done using gradients of compressed weights through backpropagation – a method not feasible for large language models (LLMs). Instead, we have adopted a strategy inspired by Intel's Incremental Network Quantization, which employs a progressive quantization strategy[4]. We merge linear layer weights with adaptors, compress them, and then freeze them. The compression loss is compensated by adaptors in other uncompressed layers. This effectively encodes LoRA's information into the codebook without necessitating additional memory.
>
> ---
> A4: At this juncture, with a compression ratio of 95%, our method collapses due to substantial compression errors, resulting in the compressed model yielding garbled inference outcomes, which fail to reflect the impact of salience on model compression.
>
> ---
> Reference:
>
> [1] Lin J, Tang J, Tang H, et al. AWQ: Activation-aware Weight Quantization for LLM Compression and Acceleration[J]. arXiv preprint arXiv:2306.00978, 2023.
>
> [2] Kim S, Hooper C, Gholami A, et al. SqueezeLLM: Dense-and-Sparse Quantization[J]. arXiv preprint arXiv:2306.07629, 2023.
>
> [3] Shao W, Chen M, Zhang Z, et al. Omniquant: Omnidirectionally calibrated quantization for large language models[J]. arXiv preprint arXiv:2308.13137, 2023.
>
> [4] Zhou A, Yao A, Guo Y, et al. Incremental network quantization: Towards lossless cnns with low-precision weights[J]. arXiv preprint arXiv:1702.03044, 2017.

---

> ### Author Response · Authors · 2023-11-22
> **Official Comment 2**
>
> ### Table 1: Comparative analyses focusing on the sources of data and the number of samples used. ‘Recovery’ indicates whether LLM-codebooks use the recovery stage. (𝐝,𝐤) represents each codebook has 𝐤 codewords of dimension 𝐝.
>
> | Method | Recovery | Dataset | Sample size | Wiki2↓ | PTB↓ |
> |--------|----------|---------|-------------|--------|------|
> | LLaMA-7B | × | - | - | 12.6 | 53.8 |
> | (𝐝,𝐤)=(4,2^12) | × | Bookcorpus | 50 | 13.61 | 57.01 |
> | (𝐝,𝐤)=(4,2^12) | × | Wikitext2 | 50 | 13.84 | 57.70 |
> | (𝐝,𝐤)=(4,2^12) | × | C4 | 50 | 14.14 | 59.79 |
> | (𝐝,𝐤)=(4,2^12) | × | Bookcorpus | 15 | 13.65 | 58.15 |
> | (𝐝,𝐤)=(4,2^12) | × | Bookcorpus | 50 | 13.61 | 57.01 |
> | (𝐝,𝐤)=(4,2^12) | × | Bookcorpus | 100 | 13.59 | 56.11 |
>
>
> ---
> ### Table 2: Ablation experiments by omitting the use of salience. ‘Recovery’ indicates whether LLM-codebooks use the recovery stage. (𝐝,𝐤) represents each codebook has 𝐤 codewords of dimension 𝐝.
>
> | Method                                       | Recovery | Cluster mode     | Rank | Learnable Params | Wiki2↓ | PTB↓   |
> |----------------------------------------------|----------|------------------|------|------------------|--------|--------|
> | LLaMA-7B                                     | ×        | -                | -    | -                | 12.6   | 53.8   |
> | (𝐝,𝐤)=(4,2^12)                               | ✓        | Common           | 64   | 2.32%            | 15.14  | 59.96  |
> | (𝐝,𝐤)=(4,2^12)                               | ✓        | Common           | 128  | 4.53%            | 15.11  | 59.73  |
> | (𝐝,𝐤)=(4,2^12)                               | ✓        | Common           | 256  | 8.67%            | 15.10  | 59.66  |
> | (𝐝,𝐤)=(4,2^12)                               | ✓        | Hessian-aware    | 64   | 2.32%            | 13.55  | 55.87  |
>
>
> ---
> ### Table 3:  Zero-shot performance of the compressed LLaMA-7B and LLaMA-13B. ‘rec’ indicates whether LLM-codebooks use the recovery stage. (𝐝,𝐤) represents each codebook has 𝐤 codewords of dimension 𝐝.
>
> | Ratio  | Method | Wiki2↓ (LLaMA-7B) | PTB↓ (LLaMA-7B) | Wiki2↓ (LLaMA-13B) | PTB↓ (LLaMA-13B) |
> | ------ | ------ | --------------------------- | -------------------------- | ----------------------------- | ---------------------------- |
> | 0.0%   | LLaMA-7B | 12.6 | 53.8 | 11.58 | 44.56 |
> | 73.3%  | SqueezeLLM-4bit | 12.82 | 54.81 | 11.73 | **45.04** |
> | 73.3%  | AWQ-128g-4bit | 12.96 | 56.56 | 11.94 | 45.05 |
> | 73.3%  | OmniQ-128g-4bit | 12.87 | 54.93 | **11.71** | 45.73 |
> | 73.9%  | (𝐝,𝐤)=(4,2^15) | 12.82 | 54.81 | 11.95 | 45.70 |
> | 73.9%  | (𝐝,𝐤)=(4,2^15)-rec | **12.81** | **54.78** | 11.94 | 45.68 |
> | 79.7%  | SqueezeLLM-3bit | 13.91 | 57.22 | 12.75 | 46.93 |
> | 79.7%  | AWQ-128g-3bit | 14.89 | 59.59 | 12.92 | 48.35 |
> | 79.7%  | OmniQ-128g-3bit | 13.78 | 57.67 | **12.36** | 48.27 |
> | 80.0%  | (𝐝,𝐤)=(4,2^12) | 13.59 | 56.11 | 12.62 | 47.16 |
> | 80.0%  | (𝐝,𝐤)=(4,2^12)-rec | **13.55** | **55.87** | 12.59 | **46.90** |
> | 85.8%  | OmniQ-128g-2bit | 22.73 | 111.24 | 17.88 | 72.75 |
> | 86.1%  | (𝐝,𝐤)=(8,2^15) | 23.48 | 88.97 | 18.50 | 67.42 |
> | 86.1%  | (𝐝,𝐤)=(8,2^15)-rec | **19.92** | **75.21** | **17.18** | **62.22** |

---

### Official Review · Reviewer_1Uyx · 2023-10-30

**Soundness:** 3 good
**Presentation:** 3 good
**Contribution:** 3 good
**Rating:** 5
**Confidence:** 5

**Summary:**

This paper introduces LLM-codebook for extreme compression of LLMs, which clusters LLM weights into codebooks (in KB) with three stages: (i) the salience stage derives the salience of the random layer's weight through the hessian matrix; (ii) the cluster stage employs the hessian-aware k-means algorithm to cluster the codebook, and (iii) the recover stage uses LoRA for performance recovery. The paper conducts experiments on Llama-7b and vicuna-7b, and compares with both pruning and quantization baselines. The results demonstrate the superiority of LLM-codebook in achieving higher compression ratios.

**Strengths:**

1. It achieves higher compression ratios for LLMs.
2. This paper refrains from viewing pruning and quantization as two distinct paths for LLM compression. Instead, it perceives them as techniques for information compression[1]. Consequently, the paper's narrative does not adhere to the existing quantization or pruning pipeline. It introduces a unique compression technique: compressing LLM weights by clustering them and storing them in kilobyte-scale codebooks. This perspective is novel.

**Weaknesses:**

1. This work is essentially an application of production quantization on LLMs. Although the final performance surpasses the baselines, the method itself does not present much novelty. The idea of Hessian-aware k-means has also been utilized in previous works [1][2]
2. From the perspective of LLM compression ratio, this work does not compare with the state-of-the-art quantization works [2][3][4]
3. The current evaluation is focused solely on the 7b model. Could the author also provide evaluations on larger models, such as Llama-13b?
[1] TOWARDS THE LIMIT OF NETWORK QUANTIZATION ICLR 2017, SAMSUNG
[2] https://arxiv.org/abs/2306.07629
[3] https://arxiv.org/abs/2307.13304
[4] https://arxiv.org/pdf/2306.00978.pdf

**Questions:**

Please refer to the weaknesses section

---

> ### Author Response · Authors · 2023-11-22
> **Official Comment 1**
>
> Dear reviewer,
>
> I apologize for the delayed response. Your insightful feedback has been immensely valuable in guiding the refinement of our work. Despite the constraints of limited laboratory resources, I have endeavored to conduct as many supplementary experiments as possible to address your comments comprehensively.
>
> ---
> The updated part of our method:
>
> Inspired by the approaches of AWQ[1] and SqueezeLLM[2] in preserving important weights, we upgrade Hessian-aware K-means algorithm to retain the weight and position of the salience value in the first 0.66% in a compressed sparse row. This significantly enhances the compression performance with a 2% reduction in the compression ratio. The preserved weights are excluded from the Hessian-aware k-means clustering, which also reduces clustering errors.
>
> ---
> A1: We sincerely apologize for our insufficient research, as we failed to recognize that the concept of Hessian-aware k-means has already been utilized in previous works [3][4]. Our implementation of Hessian-aware k-means differs in several key aspects. First, by performing a Taylor expansion on the loss function, we obtain three terms: the first related to the gradient, the second to the Hessian matrix, and the third being a redundant term. Previous works often omit the first term, assuming gradients are close to zero. However, in our case, since the dataset used for computing gradients and Hessian matrices is not extracted from the original training set, gradients are not negligible and should be considered. We regret that our naming convention may have led to the misconception that the significance of weights is solely determined by Hessian calculations. As shown in Table1 of Official Comment 2, we conducted experiments showing that considering both the first and second terms is more effective than considering them separately. Additionally, in our salience-based centroid weighting update, we noticed that many weights have salience values close to zero, leading to significant compression errors. To mitigate this, we added salience.max() to all salience values for smoothing.
>
> Furthermore, our codebook-based LoRA approach in the recovery stage marks a significant innovation. This method deviates from earlier practices in CNN-related research, where updating the codebook was done using gradients of compressed weights through backpropagation – a method not feasible for large language models (LLMs). Instead, we have adopted a strategy inspired by Intel's Incremental Network Quantization, which employs a progressive quantization strategy[5]. We merge linear layer weights with adaptors, compress them, and then freeze them. The compression loss is compensated by adaptors in other uncompressed layers. This effectively encodes LoRA's information into the codebook without necessitating additional memory.
>
> ---
> A2&A3:
> In Table 2 of Official Comment 2, we have supplemented the low-bit compression comparison with Omniquant[6], SqueezeLLM, and AWQ, demonstrating that our work achieves optimal performance in 2/3-bit compression of LLaMA-7b and LLaMA-13B. Although QuIP represents an excellent piece of work, the absence of open-source code of LLaMA precludes comparison. As shown in Table 3 of Official Comment 2, in exploring the potential of combining low-bit compression with structural clustering for further compression, we conducted experiments with Codebook PTQ, indicating that the results for W8A16 and W4A16 are very close to those of W16A16.
>
> ---
> Reference:
>
> [1] Lin J, Tang J, Tang H, et al. AWQ: Activation-aware Weight Quantization for LLM Compression and Acceleration[J]. arXiv preprint arXiv:2306.00978, 2023.
>
> [2] Kim S, Hooper C, Gholami A, et al. SqueezeLLM: Dense-and-Sparse Quantization[J]. arXiv preprint arXiv:2306.07629, 2023.
>
> [3] TOWARDS THE LIMIT OF NETWORK QUANTIZATION ICLR 2017, SAMSUNG
>
> [4] https://arxiv.org/abs/2306.07629
>
> [5] Zhou A, Yao A, Guo Y, et al. Incremental network quantization: Towards lossless cnns with low-precision weights[J]. arXiv preprint arXiv:1702.03044, 2017.
>
> [6] Shao W, Chen M, Zhang Z, et al. Omniquant: Omnidirectionally calibrated quantization for large language models[J]. arXiv preprint arXiv:2308.13137, 2023.

---

> ### Author Response · Authors · 2023-11-22
> **Official Comment 2**
>
> ### Table 1: Comparation of calculation mode of weight salience. 'First term' refers to the calculation of salience utilizing the first term (related to the gradient) in the Taylor expansion of the loss function. 'Second term' denotes the use of the second term (related to the Hessian matrix) for this purpose. 'Mix' represents the addition of the first and second terms. (𝐝,𝐤) represents each codebook has 𝐤 codewords of dimension 𝐝. ‘Recovery’ indicates whether LLM-codebooks use the recovery stage.
>
> | Method | Recovery | Ratio | Salience mode | Wiki2↓ | PTB↓ |
> |--------|----------|-------|---------------|--------|------|
> | LLaMA-7B | × | 0.0% | - | 12.6 | 53.8 |
> | (𝐝,𝐤) = (4, 2^15) | × | 73.9% | First term | 12.85 | 55.25 |
> | (𝐝,𝐤) = (4, 2^15) | × | 73.9% | Second term | 13.30 | 55.79 |
> | (𝐝,𝐤) = (4, 2^15) | × | 73.9% | Mix | **12.82** | **54.81** |
> | (𝐝,𝐤) = (4, 2^12) | × | 80.0% | First term | 13.61 | 56.52 |
> | (𝐝,𝐤) = (4, 2^12) | × | 80.0% | Second term | 14.05 | 57.06 |
> | (𝐝,𝐤) = (4, 2^12) | × | 80.0% | Mix | **13.59** | **56.11** |
> | (𝐝,𝐤) = (8, 2^15) | × | 86.1% | First term | 23.57 | 90.64 |
> | (𝐝,𝐤) = (8, 2^15) | × | 86.1% | Second term | 24.42 | 91.05 |
> | (𝐝,𝐤) = (8, 2^15) | × | 86.1% | Mix | **23.48** | **88.97** |
>
> ---
> ### Table 2: Zero-shot performance of the compressed LLaMA-7B and LLaMA-13B. ‘rec’ indicates whether LLM-codebooks use the recovery stage. (𝐝,𝐤) represents each codebook has 𝐤 codewords of dimension 𝐝.
>
> | Ratio  | Method | Wiki2↓ (LLaMA-7B) | PTB↓ (LLaMA-7B) | Wiki2↓ (LLaMA-13B) | PTB↓ (LLaMA-13B) |
> | ------ | ------ | --------------------------- | -------------------------- | ----------------------------- | ---------------------------- |
> | 0.0%   | LLaMA-7B | 12.6 | 53.8 | 11.58 | 44.56 |
> | 73.3%  | SqueezeLLM-4bit | 12.82 | 54.81 | 11.73 | **45.04** |
> | 73.3%  | AWQ-128g-4bit | 12.96 | 56.56 | 11.94 | 45.05 |
> | 73.3%  | OmniQ-128g-4bit | 12.87 | 54.93 | **11.71** | 45.73 |
> | 73.9%  | (𝐝,𝐤)=(4,2^15) | 12.82 | 54.81 | 11.95 | 45.70 |
> | 73.9%  | (𝐝,𝐤)=(4,2^15)-rec | **12.81** | **54.78** | 11.94 | 45.68 |
> | 79.7%  | SqueezeLLM-3bit | 13.91 | 57.22 | 12.75 | 46.93 |
> | 79.7%  | AWQ-128g-3bit | 14.89 | 59.59 | 12.92 | 48.35 |
> | 79.7%  | OmniQ-128g-3bit | 13.78 | 57.67 | **12.36** | 48.27 |
> | 80.0%  | (𝐝,𝐤)=(4,2^12) | 13.59 | 56.11 | 12.62 | 47.16 |
> | 80.0%  | (𝐝,𝐤)=(4,2^12)-rec | **13.55** | **55.87** | 12.59 | **46.90** |
> | 85.8%  | OmniQ-128g-2bit | 22.73 | 111.24 | 17.88 | 72.75 |
> | 86.1%  | (𝐝,𝐤)=(8,2^15) | 23.48 | 88.97 | 18.50 | 67.42 |
> | 86.1%  | (𝐝,𝐤)=(8,2^15)-rec | **19.92** | **75.21** | **17.18** | **62.22** |
>
> ---
> ### Table 3: Combining low-bit post-training quantization (PTQ) with LLM-Codebook without recovery stage. ‘g128’ represents PTQ method with a group size of 128 is used for all the codebooks. (𝐝,𝐤) represents each codebook has 𝐤 codewords of dimension 𝐝.
> | Method | Recovery | Bits | Wiki2↓ (LLaMA-7B) | PTB↓ (LLaMA-7B) | Wiki2↓ (LLaMA-13B) | PTB↓ (LLaMA-13B) |
> | ------ | -------- | ---- | --------------------------- | --------------------------- | ---------------------------- | ---------------------------- |
> | (𝐝,𝐤)=(4,2^15)  | × | W16A16 | 12.82 | 54.81 | 11.95 | 45.70 |
> | (𝐝,𝐤)=(4,2^15), PTQ-g128 | × | W8A16 | 12.82 | 54.81 | 11.95 | 45.70 |
> | (𝐝,𝐤)=(4,2^15), PTQ-g128 | × | W4A16 | 13.46 | 57.22 | 11.95 | 45.70 |
> | (𝐝,𝐤)=(4,2^15), PTQ-g128 | × | W3A16 | 16.75 | 67.16 | 14.33 | 53.54 |
> | (𝐝,𝐤)=(4,2^12)  | × | W16A16 | 13.59 | 56.11 | 12.62 | 47.16 |
> | (𝐝,𝐤)=(4,2^12), PTQ-g128 | × | W8A16 | 13.59 | 56.11 | 12.62 | 47.16 |
> | (𝐝,𝐤)=(4,2^12), PTQ-g128 | × | W4A16 | 13.59 | 56.11 | 12.62 | 47.16 |
> | (𝐝,𝐤)=(4,2^12), PTQ-g128 | × | W3A16 | 18.11 | 73.18 | 15.37 | 56.55 |
> | (𝐝,𝐤)=(8,2^15) | × | W16A16 | 23.48 | 88.97 | 18.50 | 67.42 |
> | (𝐝,𝐤)=(8,2^15), PTQ-g128 | × | W8A16 | 23.48 | 88.97 | 18.50 | 67.42 |
> | (𝐝,𝐤)=(8,2^15), PTQ-g128 | × | W4A16 | 23.48 | 88.97 | 19.93 | 73.18 |
> | (𝐝,𝐤)=(8,2^15), PTQ-g128 | × | W3A16 | 44.39 | 192.81 | 27.78 | 109.86 |

---

### Official Review · Reviewer_LrZH · 2023-12-03

**Soundness:** 3 good
**Presentation:** 3 good
**Contribution:** 2 fair
**Rating:** 5
**Confidence:** 5

**Summary:**

This paper proposes a codebook-based compression method for LLM. The compression technique indeed has a higher potential to achieve higher accuracy than the naive uniform quantization. However, the idea is not new and has been widely used in model compression before the era of LLM. Besides, the LoRA introduced to recover the accuracy is also not new. Therefore, the novelty of this paper is limited.

**Strengths:**

- The paper is well-written and clear
- The improvement compared with the naive quantization method is solid.

**Weaknesses:**

- The codebook-based method is not new and widely used the traditional compression methods.
- The LoRA used to recover the accuracy is not first proposed in this paper.
- The experiment should include large models.
- The latency of the decompression with a codebook should be fully discussed.

**Questions:**

Please refer to the part of weakness.

---

### Meta-Review · Area_Chair_4pac · 2023-12-10

**Metareview:**

This paper presents "a novel Hessian-aware K-means algorithm" to quantize LLMs. The results of the compression of Llama 7B and Vicuna 7B are convincing with a strong reduction in side (6x) with no loss of downstream tasks performance. The issues with the paper are that 1. the codebook-index method is basically product quantization (PQ, Jegou et al. 2010) which has been applied several times in model compression (e.g. QuantNoise, Fan et al. 2021), 2. LoRa-based tuning is necessary (compared to some other methods this is being compared to, e.g. GPTQ). The authors rebuttal is positive, but for instance not all of reviewer 1Uyx's comments are correctly addressed nor was the paper updated significantly. In its current shape, the paper is not suitable for publication at ICLR. This is still a promising paper and research direction, but the writing and framing needs to be improved, and more experimental evidence (or an open source library for others to quantize their models with it and benchmark themselves) would help convince reviewers.

**Justification For Why Not Higher Score:**

Simple method, not novel, good results, paper that omits somewhat that that this method and domain existed.

**Justification For Why Not Lower Score:**

N/A

---

### Decision · Program_Chairs · 2024-01-16

Reject